# Effects of El Niño/La Niña on the Number of Imported Shigellosis Cases in the Republic of Korea, 2004–2017

**DOI:** 10.3390/ijerph18010211

**Published:** 2020-12-30

**Authors:** Jong-Hun Kim, Jisun Sung, Ho-Jang Kwon, Hae-Kwan Cheong

**Affiliations:** 1Department of Social and Preventive Medicine, Sungkyunkwan University School of Medicine, Suwon 16419, Korea; kimjh32@skku.edu (J.-H.K.); vivid82th@hanmail.net (J.S.); 2Department of Preventive Medicine, Dankook University College of Medicine, Cheonan 31116, Korea; hojangkwon@gmail.com

**Keywords:** shigellosis, bacterial dysentery, South Korea, Republic of Korea, El Niño, La Niña, ENSO, IOD, Southeast Asia

## Abstract

Shigellosis is a major diarrheal disease in low- and middle-income countries. Although the incidence of such diseases in South and Southeast Asia has been associated with climate fluctuations linked to the El Niño–Southern Oscillation (ENSO), the impact of ENSO on shigellosis infections remains unknown. Data reported to being infected with shigellosis while traveling abroad from 2004 to 2017 were obtained from the Korea Centers for Disease Control and Prevention. We investigated the relationship between the Oceanic Niño Index (ONI) and Indian Ocean Dipole Mode Index and the relative risk of shigellosis in outbound travelers using distributed lag linear and non-linear models. From 2004 to 2017, 87.1% of imported shigellosis was infected in South and Southeast Asian countries. The relative risk of imported shigellosis infection in outbound travelers increased as the ONI decreased. In the association with the five-month cumulative ONI, the relative risk of infection continuously increased as the La Niña index gained strength. Climate fluctuations associated with the La Niña phenomenon in South and Southeast Asian countries can lead to issues in sanitation and water safety. Our findings suggest that the decreasing trend in the ONI is associated with an increased incidence of shigellosis in these countries.

## 1. Introduction

Shigellosis is a clinical syndrome with symptoms such as nausea, vomiting, diarrhea, and abdominal cramps caused by the Shigella species invading intestinal epithelial cells [1,2]. Shigella is an important etiologic agent of travelers’ diarrhea and is only carried by humans and upper primates [3]. It is mainly transmitted between people through the fecal–oral route and also through the intake of contaminated food or water. According to research conducted in South and Southeast Asian countries, the annual incidence of shigellosis from 2000 to 2004 for all ages was 0.6–7.9 cases per 1000 residents [4]. Shigella has been reported to account for 2.2% to 8.0% of travelers’ diarrhea cases in South and Southeast Asian countries [5,6].

Diarrheal diseases, including shigellosis, are well known as climate-sensitive diseases; their incidence usually increases with the occurrence of drought or floods. This is related to poor hygiene, sanitation, and a lack of clean drinking water. Weather events such as droughts or floods in South and Southeast Asian countries are affected by the El Niño Southern Oscillation (ENSO) phenomenon, which refers to variations in climate conditions resulting from the interaction of sea surface temperatures and atmospheric pressure in the tropical Pacific [7,8]. El Niño, commonly called “warm event,” and La Niña, called “cool event,” are parts of the ENSO cycle that last 12–18 months over a period of 2–7 years and the impact of which depends on geographical characteristics. Many studies have demonstrated the association between the ENSO and diarrheal diseases, particularly cholera [9,10,11,12]. A few studies have investigated the link between shigellosis and ENSO. A study conducted in Bangladesh showed that the pattern of shigellosis outbreaks was associated with the monsoon floods in a manner similar to that of cholera, and the floods were related to ENSO activity from the preceding winter [13]. A Chinese study that assessed the impact of the ENSO on shigellosis reported a negative association between the number of monthly shigellosis cases and monthly Southern Oscillation Index, which is an index of the ENSO [14]. A significant association between La Niña-related floods and an increased risk of shigellosis infection has also been reported in Peru [15]. Although studies have reported the relationship between the incidence of shigellosis in individual countries and climate indices, reviews from South and Southeast Asian regions have been insufficient. This study aimed to investigate the impact of the ENSO on imported shigellosis cases among South Koreans who had traveled to South and Southeast Asian countries.

## 2. Materials and Methods

### 2.1. Data Sources: Shigellosis Cases and Outbound Travellers

We analyzed the relationship between the number of imported shigellosis cases from South and Southeast Asian countries and sea surface temperature indices as climate indices after adjusting for the number of outbound South Korean travelers, seasonality, and long-term time trends. In South Korea, shigellosis is designated as a notifiable infectious disease that must be reported to the national infectious disease reporting system within 24 h. Legal infectious diseases are reported by both the doctor who diagnosed them and the institutions that confirmed the pathogens of infectious diseases. Once a case of notifiable infectious disease is reported, a detailed epidemiological investigation is conducted by health authorities on the risk factors at the time of infection. The epidemiologic surveillance data of patients with shigellosis were obtained from the Korea Centers for Disease Control & Prevention (KCDC). Among outbound travelers, we included only those individuals with shigellosis infections acquired after traveling to South and Southeast Asian countries (South Asian countries: India, Nepal, Bangladesh, Pakistan; Southeast Asian Countries: Cambodia, Philippines, Vietnam, Thailand, Indonesia, Laos, Myanmar, Malaysia, Singapore). The monthly statistical data of overseas South Korean tourists provided by the Korea Tourism Organization were used to determine the number of outbound travelers [16].

### 2.2. Data Sources: Climate Indices

For the tropical Pacific and Indian Oceans, there are modes of internal fluctuations leading to climate fluctuations, namely, the ENSO and the Indian Ocean Dipole (IOD), respectively. The ENSO is a climatic event in the Pacific that has a wide impact on the world’s climate and is particularly associated with droughts and floods. Meanwhile, IOD is associated with sea-level temperature fluctuations in the Indian Ocean as well as droughts and abnormal rainfall in Southeast Asia and the eastern African region. We included the dipole mode event in Indian Ocean as an independent variable because it is independent of the ENSO in the Pacific. The Oceanic Niño Index (ONI) is a major index used to monitor the ENSO. The ONI is calculated by a three-month time averaging of the sea surface temperature anomalies in an area of the east-central equatorial Pacific Ocean called the Nino 3.4 region (5S to 5N; 170W to 120W). Just as ENSO-related climatic fluctuations are presented using ONI indicators, the Indian Ocean Dipole Mode Index (DMI) is used to present the IOD-related climatic fluctuations. DMI is an indicator of sustained changes in the difference between sea surface temperatures of the tropical western (50° E to70° E, 10° S to 10° N) and eastern (90° E to 110° E, 10° S to the equator) Indian Ocean. We used monthly data on the ONI and DMI provided by the Asia-Pacific Economic Cooperation (APEC) climate center [17].

### 2.3. Data Analysis

After identifying trends in domestic and imported cases of shigellosis reported in South Korea from 2004 to 2017, we classified imported cases by the countries where transmission of the infection was estimated to have occurred. The country in which the transmission of such shigellosis infection is estimated to have occurred was based on the records of the epidemiological investigation. In this study, distributed lag non-linear models (DLNM) and Poisson regression models were used to analyze non-linear exposure-response relationships while considering the lag-time effects of climate indices for imported shigellosis cases [18,19]. The dependent variable is the number of monthly shigellosis patients reported to be infected during travel to South and Southeast Asia, and the independent variable was the monthly ONI. The monthly number of overseas travelers, monthly DMI, seasonality, and long-term time trends were the variables adjusted in the models. As the occurrence of El Niño is based on the ONI, we attempted to interpret the results based on this index. Analyses were performed using the R package “dlnm”.
(1)$$Y_{t}=Quasi-poisson\left(Y_{t}\right),\;t=1,\ldots \ldots ,n$$
(2)$$\begin{array}{c} \log\left(Y_{t}\right)=\;\alpha +\sum_{l=1}^{L}\beta_{0}\left(ONI_{t,l}\right)+\sum_{l=1}^{L}\beta_{1}\left(DMI_{t,l}\right)+\beta_{2}\times ns\left(season,\lambda_{1}\right)\\ +\beta_{3}\times ns\left(time,\lambda_{2}\right)+\beta_{4}\times \log\left(N_{t}\right) \end{array}$$
where *t* is the month reported to the KCDC; *Y_t_* is the monthly imported number of shigellosis cases for month *t*; α is the intercept; *ONI_t,l_*(*DMI_t_*_,*l*_) is the matrix obtained by applying the DLNM to the ONI(DMI); *l* is the lag month; *L* is the maximum lag; *N_t_* is an offset to control for the number of outbound travelers; *ns*(s*eason*, *λ*_1_) is the natural cubic spline smoothing function of the ‘season’ variable for the control of seasonal patterns; *ns*(*time*, *λ*_2_) is the natural cubic spline smoothing function of the ‘time’ variable for the control of long-term time trends and fluctuations; *β*_0_, *β*_1_, *β*_2_, *β*_3_, and *β*_4_ are the coefficients for *ONI_t,l_*, *DMI_t,l_*, *ns*(*season*, *λ*), and log(*N_t_*), respectively. We used a natural cubic spline basis with three degrees of freedom (df) for the ONI and two df for the lag. A natural cubic spline with 3 df was used for both the DMI and the lag. To control seasonal patterns and long-term time trends, natural cubic splines with 3 df and 7 df were used, respectively. To choose the best model with optimal df, we adopted the smaller Akaike Information Criterion (AIC) value for the quasi-Poisson model and conducted a sensitivity analysis (Appendix A).

### 2.4. Ethical Statement

This study was approved by the institutional review board of Dankook University (IRB No. DKUH 2018-09-011).

## 3. Results

The overall number of shigellosis cases reported in South Korea tended to decrease from 2004 to 2017 (Figure 1). The total number of cases, which occurred close to 500 cases in 2004, has decreased significantly since 2014 and is reported to be less than 120 cases. In the classification by country suspected of being infected, we observed a gradual decrease in the proportion of domestic cases, while the proportion of cases imported from abroad increased progressively. From 2004 to 2017, a total of 698 imported shigellosis cases were reported to the KCDC (Table 1). When assessing the overseas regions suspected of originating imported infections, 476 cases (68.2%) were estimated as being infected with shigellosis while traveling in Southeast Asian countries. The next frequent regions were; 132 cases (18.9%) in South Asian countries, 63 cases (9.0%) in East Asian countries, and 27 cases (3.8%) in other regions. More specifically, in Southeast Asian countries, 137 cases (19.6%) were reported among those who travelled to Cambodia, followed by the Philippines with 119 cases (17.0%) and Vietnam with 89 cases (12.8%). A total of 132 cases were reported by travelers who visited South Asian countries, with 124 cases (17.8%) being infected during travel to India, accounting for most cases. In East Asia, it was the largest in China, with 48 cases (6.9%). From 2004 to 2017, a total of four El Niño and five La Niña events occurred (El Niño: June 2004–May 2005, August 2006–January 2007, June 2009–April 2010, and March 2015–May 2016; La Niña: September 2007–May 2008, July 2010–April 2011, August 2011–March 2012, August 2016–January 2017, and September 2017–December 2017) (Figure 2).

The number of outbound travelers in South Korea averaged about 750,000 per month in 2004 but continued to increase, exceeding an average of 1.1 million per month in 2007. In 2008–2009, the number of outbound travelers decreased due to the global economic crisis but has steadily increased since then, surpassing an average of 2 million per month in 2017. In the scatter plot between the ONI (DMI) and the number of imported shigellosis cases at lag 0 month from South and Southeast Asian countries, the number of cases tended to increase as the ONI decreased. However, we observed no distinct difference in the DMI values (Figure 3).

As a result of analyzing the relative risk of the ONI and number of shigellosis cases by classifying from zero months to five months, the relative risk associated with shigellosis infection increased as the ONI decreased, until lagged two months (Figure 4). When 0 (ONI) was used as a reference value, the relative risk at −0.5 (ONI) for lag 0 month was 1.25 (1.12–1.40) and that at −0.5 (ONI) for lag 1, 2, and 3 months was 1.17 (1.08–1.27), 1.09 (1.03–1.16), and 1.02 (0.97–1.08), respectively. In the five-month cumulative association, the relative risk was 1.38 (1.01–1.89) at −0.5 (ONI), 2.57 (1.46–4.53) at −1.0 (ONI), and 3.46 (1.64–7.33) at −1.2 (ONI) (Figure 5). The relative risk increased as the five-month cumulative was stronger (decreased). The relative risk increased significantly at index values below −0.5 (ONI), however, there was no significant change observed at index values above −0.5 (ONI).

## 4. Discussion

We reported that the risk of infection with shigellosis in South Korean travelers returning from South and Southeast Asian countries is influenced by the ONI, a large-scale oceanic index. The risk of this infection was non-linear and markedly increased in the La Niña situation, where the overall cumulative ONI level was less than −0.5 for a period of 5 months. El Niño (La Niña) is based on a three-month moving average of the deviation in sea surface temperatures in the monitoring area (Nino 3.4 region) for El Niño (La Niña). The first month of the period when the ONI of 0.5 °C or higher lasts more than five months is considered the starting point for El Niño. Additionally, the first month of the period when the ONI of 0.5 °C or lower lasts more than five months is considered the starting point for La Niña. Based on this definition, our study analyzed the overall cumulative association from the present to the last five months. The influence of meteorological factors on the incidence of shigellosis has a lagged effect, and it is known that a time lag of about 0–2 months considering the survival and incubation period of the pathogen. In our study, the time lag for ONI was classified from zero months to five months. At −0.5 (ONI), the relative risk associated with shigellosis infection was significant from lag zero months to lag two months. As the time lag was shortened, the relative risk increased. The relative risk is highest as 1.25 (1.12–1.40) at −0.5 (ONI) for lag 0 months. These results can be inferred by reflecting the temporal characteristics that the pathogen Shigella can survive in water for about four weeks, and the incubation period of shigellosis is within one week. Besides, the lagged effect on the increase in shigellosis cases increased by extreme weather events (flood, typhoon) did not exceed one week [8]. 

ENSO and IOD are climate fluctuations in the Pacific and Atlantic Oceans but are known as important indicators of events that can affect climate globally through teleconnection of the atmosphere and oceans. While fluctuations in the ENSO are relatively influential in the context of global climate fluctuations, the IOD is known to account for approximately 12% of Indian Ocean sea surface temperature fluctuations, which mainly affect countries adjacent to the Indian Ocean [20,21]. Variation in the ENSO and IOD can cause severe flooding and droughts around the world through atmospheric remote connections [7,22]. In particular, South and Southeast Asian regions, which are located in areas impacted by fluctuations in both the ENSO and IOD, are greatly affected by the climate. El Niño and La Niña events occurring during irregular fluctuations in the ENSO also have an impact on public health—associations between hazards such as droughts and flooding, and the incidences of vector-borne diseases (dengue fever and malaria) and diarrheal diseases (cholera) have been reported [23,24]. As heavy rain is generally associated with an increased incidence of intestinal pathogens as a result of contamination of the water supply, the incidence of diarrheal disease usually peaks in the rainy season in tropical regions. In Botswana, Southern Africa, an association between flooding due to the lagged 0–5 months of La Niña’s condition and diarrhoea in children under 5 years of age has been reported [25]. In the central region of Vietnam, the incidence of shigellosis has been reported to increase in the local rainy season [26]. In low- and middle-income countries, an increase in precipitation can increase the risk of shigellosis infection by water contaminated through overflowing sewage, runoff human or animal feces from the surface, or the re-suspension of sediment [8]. In this way, an increase in overall rainfall due to La Niña in the Western Pacific region can lead to contamination of drinking water in low- and middle-income countries where infrastructure is relatively poor. Existing studies on the relationship between shigellosis and climate indices have been limited in examining the impact of the ENSO directly, as many studies have been conducted in countries not located in low-latitude regions, such as China [14]. However, in these countries, it is difficult to obtain qualified data due to the limitations in diagnostic testing and infectious disease reporting system. Therefore, outbound travelers in developed countries who visit South and Southeast Asia play the role of geo-sentinel surveillance in low-latitude regions, which highlights their value in providing high-quality sampling data in this regard [27,28,29]. This is because travelers of northern countries with well-established infectious disease reporting systems, who travel and return from Southeast Asia and South Asia can be work as a random sampling of residents of the regions, a proxy of a sentinel surveillance across the region [30].

As overall water safety and hygiene levels improved in South Korea, the number of domestic cases with shigellosis decreased. However, with the increase in the number of overseas travelers, the proportion of imported cases among the total number of cases of shigellosis continued to increase. The absolute number of imported shigellosis cases changes every year due to fluctuating climate indices; the largest number of imported shigellosis cases reported in 2010–2011 was associated with the occurrence of the strongest La Niña at that time during the study period. Of the countries visited by South Korean travelers, Thailand was the most visited country among the Southeast Asian countries, followed by the Philippines, Vietnam, and Singapore (Appendix A). However, the reported number of imported shigellosis cases was highest in Cambodia, followed by the Philippines, and Vietnam. The level of water safety, sanitation, and the prevalence of water-borne and food-borne diseases all seem to affect the risk of infection at the national level [31]. In addition, it is already known that when El Niño/La Niña occurs, drought/flood frequently occurs in Southeast Asia and South Asia region. However, in small regions or individual countries, geographic differences may cause weather events that differ from the global climatic phenomenon. Therefore, when the fluctuation in the number of imported shigellosis cases expanded from the single country level to the South and Southeast Asian regions, the relationship with the climate indices and imported shigellosis cases became more apparent. According to data on international tourist destinations from the World Tourism Organization, the number of travelers who visited South and Southeast Asian countries continued to increase from 54 million in 2004 to 148 million in 2017 [32]. As the number of international travelers visiting the region increases rapidly, it is likely that the number of international travelers with shigellosis infections will continue to increase in the future. Therefore, if we can gauge the risk of shigellosis infection due to changes in climate indices in advance, it may help travelers reduce their risk of disease by practicing better personal hygiene and paying attention to their food intake. The number of cases with shigellosis who returned to South Korea after traveling to South and Southeast Asia had a negative association with the monthly ONI. A similar relationship between imported shigellosis cases and ONI will likely be observed if outbound travelers from America and Europe who have visited South and Southeast Asian countries are considered instead. As prediction models of the monthly ONI and DMI have already been developed, it is possible to build a model that predicts the number of cases with shigellosis in South and Southeast Asia in the future. 

This study has some limitations. In this study, data reported to the legal infectious disease surveillance system were used for cases of shigellosis infection among South Koreans who entered domestic after a short overseas trip. However, if a tourist infected with shigellosis improves before returning home and does not visit a medical institution in Korea, or if he/she continues to travel after receiving treatment at the local abroad after being infected with shigellosis, data may be omitted and under-reported. The incidence of diarrheal diseases in tropical countries can increase during drought and flood periods. However, there must be an ongoing surveillance system in tropical countries to evaluate the correlation between the incidence of shigellosis and our study’s findings in each country. We could not obtain reliable shigellosis surveillance data in each country, which is another limitation of our study. As this study was conducted on South Korean individuals diagnosed with shigellosis after visiting South and Southeast Asian countries, we adjusted the total number of overseas travelers. However, we did not consider regional meteorological factors according to the local area visited. Diarrheal disease is affected not only by meteorological factors, but also by personal hygiene and infrastructural socio-economic factors such as water safety and sanitation. If the spatial resolution is further narrowed to a local area or a single nation, the results may not be the same as those of our study. However, this study attempted to explain the relationship between climate indices in a tropical region and the occurrence of shigellosis. In this study, we aimed to assess the impact of the overall cumulative association of climate indices on the incidence of shigellosis in a widespread area such as South and Southeast Asian regions, rather than in a local region with specific meteorological changes.

## 5. Conclusions

The incidence of shigellosis in travelers returning from trips to each Southeast Asian country represents the status of the shigellosis outbreak in each region. However, it is difficult to obtain the incidence data of shigellosis by tourist destinations. To solve these problems, we analyzed the relationship between the ONI and the DMI and the relative risk of shigellosis for overseas travelers. As a result, we have confirmed that the risk of imported shigellosis infection in outbound travelers increases as ONI decreases. Our results suggest that the decreasing trend of ONI, which increases the probability of La Niña occurring, is related to the increased incidence of shigellosis in these South and Southeast Asian countries. It is predicted that extreme rainfall frequency will increase in the future due to climate change. Still, it is difficult to project in advance how the amplitude and spatial pattern of climatic events such as El Niño and La Niña will occur in the distant future. However, the number of international tourist arrivals in South and Southeast Asian countries is increasing rapidly every year. Therefore, it can be expected that shigellosis cases will increase significantly among international tourists visiting this area during the La Niña period. It is recommended that guidance is required to pay attention to hand hygiene and food consumption to tourists visiting the South and Southeast Asian countries during the La Niña period.

## Figures and Tables

**Figure 1 ijerph-18-00211-f001:**
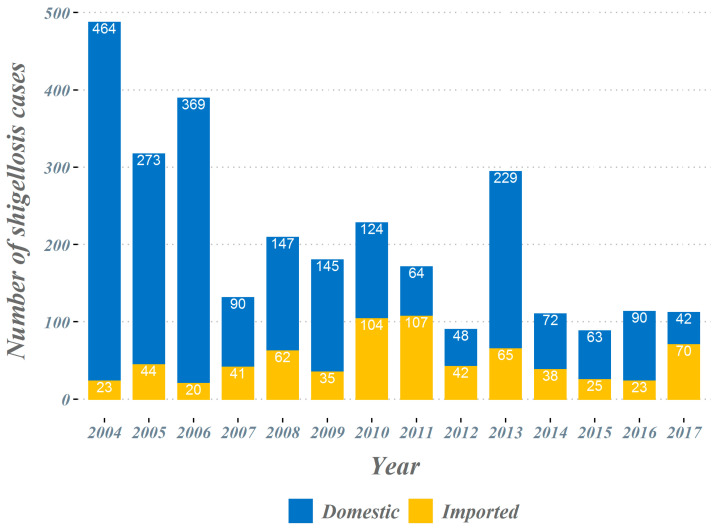
Shigellosis in the Republic of Korea, 2004–2017.

**Figure 2 ijerph-18-00211-f002:**
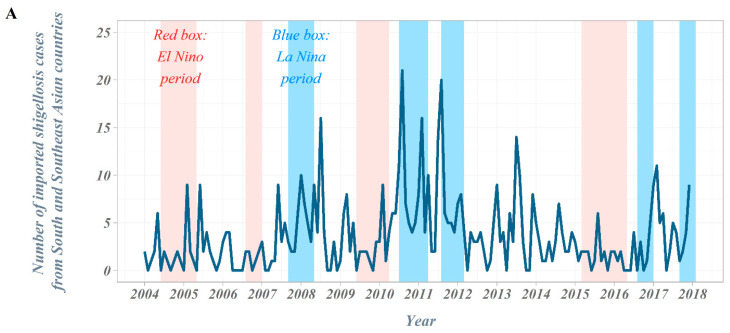
Time trends in the Republic of Korea, 2004–2017. (**A**) Number of imported shigellosis cases from South and Southeast Asian countries, 2004–2017; (**B**) time trends of climate indices: ONI and DMI, 2004–2017; (**C**) number of outbound travelers in the Republic of Korea, 2004–2017.

**Figure 3 ijerph-18-00211-f003:**
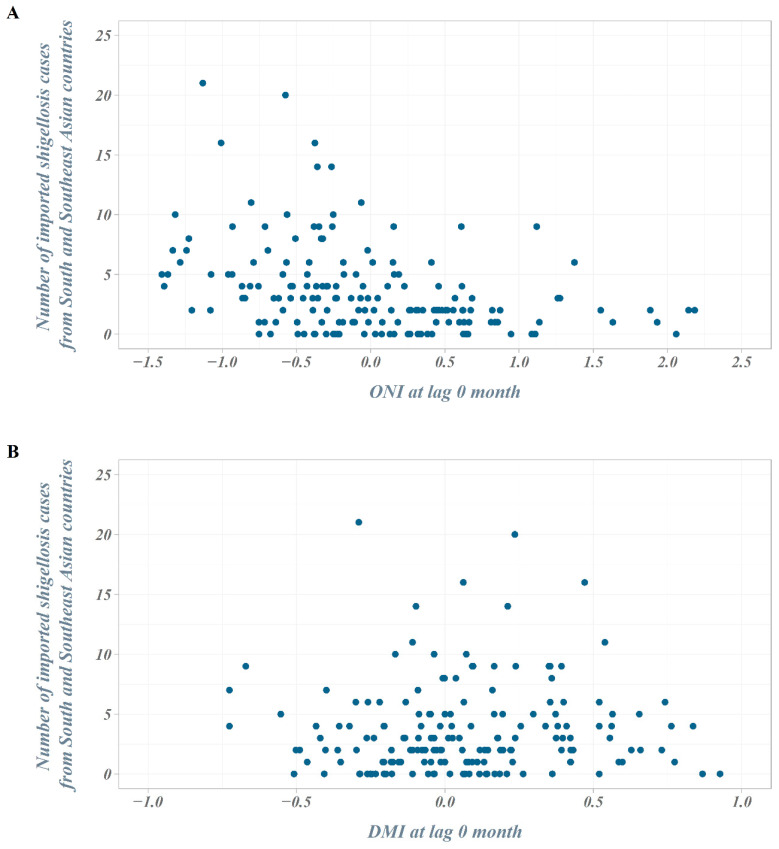
Scatter plot of the number of imported shigellosis cases and climate indices. (**A**) Scatter plot of the number of imported shigellosis cases and ONI at lag 0 month; (**B**) scatter plot of the number of imported shigellosis cases and Dipole Mode Index (DMI) at lag 0 month.

**Figure 4 ijerph-18-00211-f004:**
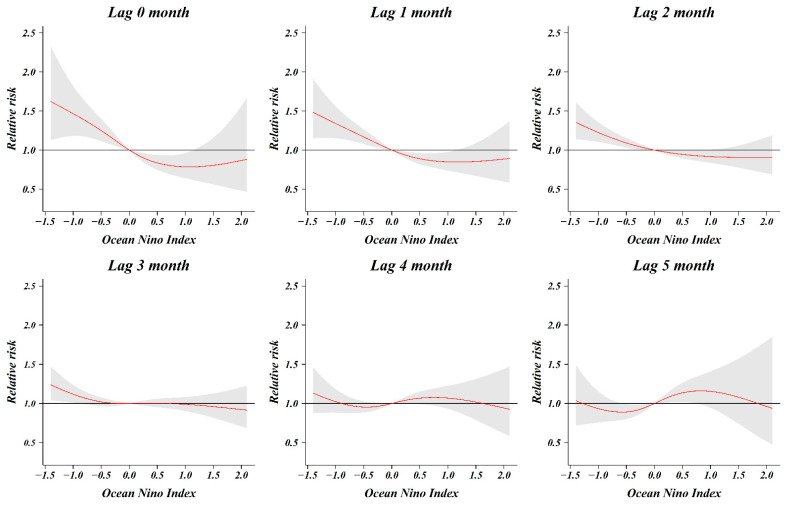
Relative risk according to the lag month of Oceanic Niño Index (ONI).

**Figure 5 ijerph-18-00211-f005:**
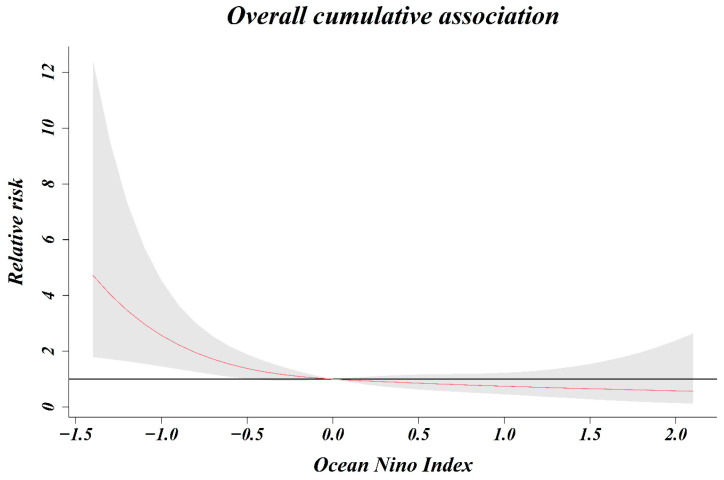
Overall cumulative association (relative risk and ONI).

**Table 1 ijerph-18-00211-t001:** Distribution of imported cases of shigellosis across travel destinations, 2004–2017.

Region	Country	Cases	%
Total		698	100.0%
Southeast Asia	Subtotal	476	68.2%
	Cambodia	137	19.6%
	Philippines	119	17.0%
	Vietnam	89	12.8%
	Thailand	52	7.4%
	Indonesia	46	6.6%
	Laos	19	2.7%
	Myanmar	5	0.7%
	Malaysia	3	0.4%
	Singapore	3	0.4%
	Others	3	0.4%
South Asia	Subtotal	132	18.9%
	India	124	17.8%
	Nepal	3	0.4%
	Bangladesh	3	0.4%
	Others	2	0.3%
East Asia	Subtotal	63	9.0%
	China	48	6.9%
	Mongolia	11	1.6%
	Japan	3	0.4%
	Others	1	0.1%
West Asia		2	0.3%
Africa		12	1.7%
Americas		9	1.3%
Oceania		2	0.3%
Europe		1	0.1%
Unknown		1	0.1%

## Data Availability

Epidemiologic surveillance data sharing is not applicable to this article. However, basic data are open to the public. Materials are provided in Korean only. This data can be found at [http://www.kdca.go.kr/npt/biz/npp/ist/bass/bassAreaStatsMain.do].

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
