# Peer review of "Effects of El Niño/La Niña on the Number of Imported Shigellosis Cases in the Republic of Korea, 2004–2017"

_ijerph, 2020, doi:10.3390/ijerph18010211_

Round 1

Reviewer 1 Report

In this paper the authors study the effects of the El Niño Southern Oscillation (ENSO) and the Indian Ocean Dipole, IOD (included in the models), on imported shigellosis cases in South Koreans after travelling to South and Southeast Asian countries.

For this propose they use the Oceanic Niño Index (ONI) as an ENSO monitoring index and they calculate relative risk (RR) by the use of distributed lag non-linear models (DLNM) and Poisson regression models considering the lag-time effects of climate indices for imported shigellosis cases. The models are adjusted by monthly number of overseas travellers, monthly DMI, seasonality  and long-term time trends.

The results show a significant association between the increase of shigellosis RR in Korean outbound travellers and the decrease in the ONI trends.

In my opinion this is a novel and interesting manuscript. The methodology is adequate and permit the authors to really obtain sound results. These results are well presented and the discussion section is correctly structured and documented.

Perhaps I missed in the article some comment or data about the importance of water-borne shigellosis versus food-borne shigellosis.

Anyway, I think that this article is clearly publishable in this journal.

Author Response

In this paper the authors study the effects of the El Niño Southern Oscillation (ENSO) and the Indian Ocean Dipole, IOD (included in the models), on imported shigellosis cases in South Koreans after travelling to South and Southeast Asian countries.

For this propose they use the Oceanic Niño Index (ONI) as an ENSO monitoring index and they calculate relative risk (RR) by the use of distributed lag non-linear models (DLNM) and Poisson regression models considering the lag-time effects of climate indices for imported shigellosis cases. The models are adjusted by monthly number of overseas travellers, monthly DMI, seasonality and long-term time trends.

The results show a significant association between the increase of shigellosis RR in Korean outbound travellers and the decrease in the ONI trends.

In my opinion this is a novel and interesting manuscript. The methodology is adequate and permit the authors to really obtain sound results. These results are well presented and the discussion section is correctly structured and documented.

Perhaps I missed in the article some comment or data about the importance of water-borne shigellosis versus food-borne shigellosis.

Anyway, I think that this article is clearly publishable in this journal.

Authors’ response: We deeply appreciate your comment. We have revised the manuscript reflecting the reviewers’ comments.

Reviewer 2 Report

In the submitted manuscript, Kim et al. analyzed the relationship between the Oceanic Niño Index (ONI) and Indian Ocean Dipole Mode Index (DMI) and risk of shigellosis in oversea travelers in South Korea using distributed lag linear and non-linear model.

The authors found the association between decreased ONI and increased incidence of shigellosis in travelers.

Results from this study will provide helpful information to the readers in the field who are interested in this topic. Here are some suggestions for authors to consider.

1. There are no figure legends in all 5 figures, which make figures difficult to understand. I would suggest the authors to add figure legends. In addition, the authors could label three figures in figure 2 as Figure 2a, b, and c and refer these figures accordingly in the maintext.

Author Response

In the submitted manuscript, Kim et al. analyzed the relationship between the Oceanic Niño Index (ONI) and Indian Ocean Dipole Mode Index (DMI) and risk of shigellosis in oversea travelers in South Korea using distributed lag linear and non-linear model.

The authors found the association between decreased ONI and increased incidence of shigellosis in travelers.

Results from this study will provide helpful information to the readers in the field who are interested in this topic. Here are some suggestions for authors to consider.

  1. There are no figure legends in all 5 figures, which make figures difficult to understand. I would suggest the authors to add figure legends. In addition, the authors could label three figures in figure 2 as Figure 2a, b, and c and refer these figures accordingly in the maintext.

Authors’ response: We added figure legends as you suggested, and we revised the figure to make it more specific in the text. Thank you for your comments.

Reviewer 3 Report

Dear authors,

Thank you for giving me the opportunity to read and review this interesting manuscript. The topic is of public health relevance, and the paper is well designed and written.

Below a number of comments and suggestions, most of them minor. A general issue is to phrase the text so that readers not familiar with climatology, the ENSO and related phenomena can easily follow the paper.

Major issues

Methods - 2.1 Data sources: shigellosis cases and outbound travellers.

  • Please add details about the population data and cases. How was the diagnosis done, was underreporting a problem, how did you identify the location of infection?

Discussion

  • The discussion is generally rich and puts the study into context. It could be improved, however, by restructuring and bringing the study’s results more into focus. Here some suggestions:
  • Structure by region, rather than mixing findings from Africa, Asia and other locations that might differ in climatic and sociodemographic aspects. The study investigates shigellosis cases in South Korean travellers, which are in many ways different in their profile from other residents in tropical/sub-tropical countries.
  • The results are not much discussed, e.g. lag-specific impacts of ONI on shigellosis. That could be added to enrichen the insights. Discuss results in relation to the literature.
  • Add a section “Strengths and Limitations” (some limitations are mentioned already).

Minor issues

Abstract:

  • “outbound travellers imported shigellosis” is a bit confusing, consider revising. Are these travellers coming or leaving Korea? This is clearer in the full text. Here use e.g. “returning travellers” or another term.

Methods – 2.2 Data sources: climate indices

  • For the non-expert, differentiating between ENSO, the Southern Oscillation Index, the Oceanic Nino Index ONI, and Indian Ocean Dipole Mode Index is difficult. Can you add one or two sentences explaining how they are related?
  • ENSO is measured by ONI, and the IOD by DMI? You state that DMI is independent of ENSO, but why was it then included in the models? Also, in Figure 2A the two seem quite correlated in their occurrence.

Methods – 2.3 Data analyses

  • P 2, line 84 “we categorised cases by the countries suspected of causing infection in the imported cases during this period.” Unclear – what is suspected to cause the infections? A country cannot be a cause.
  • 2 line 88 “dependent variable number of patients .... categorised by month” – means the number of monthly cases?

Results

  • P3, first paragraph of results: some sentences are unclear, please consider revising.
  • g. “total number of patients, which stood at 487 in 2004, has been reported to be within 110 patients”,
  • Line 119 “Southeast Asia was the highest with 476 cases…” make clear these are the number of imported cases, not the total number in that region.
  • Figure 2 and text about outbound travellers might be better presented before showing the number of shigellosis cases among returning travellers (or in the appendix).
  • P6 line 149 “La Nina” – unclear how this relates to the main independent variable ONI. I suggest to refer only to ONI in the results, and describe its relation to El Nino/La Nina patterns in the discussion. This will help readers without climatological expertise to follow the paper.

Discussion

  • 7/8 line “El Niño (La Niña) is based on a...”; P.8, line 162/163 “Nina (Nino) … 0.5°C or higher (-0.5 °C or lower)”.. Summarizing the two phenomena in this way is efficient but difficult to follow for non-climatologists. I suggest splitting the effects of Nino/Nina in two separate sentences, or merging them in some other way.
  • Same sentence: “The first month is considered the start of El Niño” - unclear what is meant, at least for non-experts.
  • 8, line 178: Botswana is in Southern Africa, not in South Africa.
  • 8 line 185 and other places: I would avoid value-loaded terms like “developing / developed countries” (although even the UN uses these expressions). But all countries are developing (hopefully). Rather use “low- and middle-income countries, high-income countries”.
  • Line 166 “…indicators of events potential climate-related impact” – grammar / unclear what is meant
  • Line 190-192: Good and important point. It might fit better at the end of the discussion or into the conclusion, maybe even in more detail.
  • Line 197 “the largest number of imported shigellosis cases reported in 2010-2011 was related to the occurrence of the strongest La Nina” – that implies causality, but could be coincidence as it is just one observation.
  • Line 203-205 “Therefore, when the fluctuation in the number of imported…” Unclear what is meant – country-specific analyses and results were not described before.

Conclusions

  • Good summary of paper. I would suggest to expand the conclusions – what do study finding imply beyond its scope? Possibly relevance in the context of future climate change and increasing tourism?

Author Response

Thank you for giving me the opportunity to read and review this interesting manuscript. The topic is of public health relevance, and the paper is well designed and written.

Below a number of comments and suggestions, most of them minor. A general issue is to phrase the text so that readers not familiar with climatology, the ENSO and related phenomena can easily follow the paper.

Major issues

Methods - 2.1 Data sources: shigellosis cases and outbound travellers.

  • Please add details about the population data and cases. How was the diagnosis done, was underreporting a problem, how did you identify the location of infection?

Authors’ response:

  • In South Korea, shigellosis is legally designated as a notifiable infectious disease that must be reported immediately. This means that if shigellosis is diagnosed at a medical institution, it must be reported to the national infectious disease reporting system within 24 hours. South Korea has a national medical insurance system, and 100% of the people are included in the medical insurance system.
  • The criterion for diagnosis is when a pathogen of shigella is identified in a sample (stool, rectal smear) or a specific gene (ipaH) is detected in a sample (stool, rectal smear) of a patient.
  • In South Korea, notifiable infectious diseases are reported by both the doctor who diagnosed them and the institutions that confirmed the pathogens of infectious diseases. Therefore, notification failure rarely occurs except for cases of not visiting domestic medical institutions after being diagnosed with shigellosis overseas. Also, if an outbound traveller has been diagnosed with shigellosis abroad but continues to travel abroad without returning, it may not be reported.
  • When a notifiable infectious disease is reported, an epidemiological investigation is conducted. In an epidemiological investigation, various exposure risk factors such as the date of initial symptom onset, travel history, and visited countries are investigated and based on this, and the epidemiological investigator determines the area that is estimated to be infected.

Discussion

  • The discussion is generally rich and puts the study into context. It could be improved, however, by restructuring and bringing the study’s results more into focus. Here some suggestions:
  • Structure by region, rather than mixing findings from Africa, Asia and other locations that might differ in climatic and sociodemographic aspects. The study investigates shigellosis cases in South Korean travellers, which are in many ways different in their profile from other residents in tropical/sub-tropical countries.

Authors’ response:

  • El Niño/La Niña is a vast scale climate phenomenon that occurs across the Pacific Ocean and its impact is global. This study investigated teleconnections between El Niño/La Niña phenomena occurring in the centre of the Pacific Ocean and outbreaks of infectious diseases on continents far away. Shigellosis is both a waterborne and foodborne infectious disease transmitted by faecal-oral route. In each country, the coverage and water quality of tap water are different from each other, and dietary habits and cultural aspects of food are different, the mode of transmission could be diverse.
  • This study does not consider national and local characteristics of both weather and disease. Unlike climatic phenomena, meteorological phenomena have a more localized pattern. Therefore, events such as localized floods or droughts may significantly impact the outbreak of shigellosis in certain countries. The purpose of this study is to investigate how the macro-climatic-indicator such as El Niño/La Niña has an association with the increase in the number of shigellosis cases in a large region centered on Southeast Asia and South Asia. Here, returning South Korean travellers from Southeast Asia and South Asia should be considered as a randomized sampling of persons as a proxy for residents in Southeast Asia and South Asia, rather than as a population group in a particular country. Besides, since it is aimed at overseas travellers from a single country or ethnic group that share similar lifestyles and cultures, it has the advantage of reducing interference from other local risk factors, excluding climate factors. This approach is one of the usual research methods to investigate the health effects of global climate change.
  • The results are not much discussed, e.g. lag-specific impacts of ONI on shigellosis. That could be added to enrichen the insights. Discuss results in relation to the literature.

Authors’ response:

  • The following paragraph has been added.
  • The influence of meteorological factors on the incidence of shigellosis has a lagged effect, and it is known that a time lag of about 0-2 months considering the survival and incubation period of the pathogen. In our study, the time lag for ONI was classified from 0 months to 5 months. At -0.5 (ONI), the relative risk associated with shigellosis infection was significant from lag 0 months to lag two months. As the time lag was shortened, the relative risk increased. The relative risk is highest as 1.25 (1.12-1.40) at -0.5 (ONI) for lag 0 months. These results can be inferred by reflecting the temporal characteristics that the pathogen Shigella can survive in water for about four weeks, and the incubation period of shigellosis is within one week. Besides, the lagged effect on the increase in shigellosis cases increased by extreme weather events (flood, typhoon) did not exceed one week. (Line 189-198)

  • Add a section “Strengths and Limitations” (some limitations are mentioned already).

Authors’ response:

  • The strengths and limitations were naturally discussed by adding the following to the existing paragraph.
  • This study has some limitations. In this study, data reported to the legal infectious disease surveillance system were used for cases of shigellosis infection among South Koreans who entered domestic after a short overseas trip. However, if a tourist infected with shigellosis improves before returning home and does not visit a medical institution in Korea, or if he/she continues to travel after receiving treatment at the local abroad after being infected with shigellosis, data may be omitted and under-reported. (Line 260-265)

Minor issues

Abstract:

  • “outbound travellers imported shigellosis” is a bit confusing, consider revising. Are these travellers coming or leaving Korea? This is clearer in the full text. Here use e.g. “returning travellers” or another term.

Authors’ response:

  • The mentioned sentence has been modified to the following sentence.
  • Data reported being infected with shigellosis while travelling abroad from 2004 to 2017
  • From 2004 to 2017, 87.1% of imported shigellosis was infected in South and Southeast Asian countries.
  • The researchers also had many of concerns about choosing that term. The exact description will be as follows—South Koreans who entered domestic after travelling abroad in the short-term. 'Outbound traveller' seems to mean more of a short-term overseas traveller departing from South Korea, and 'returning travellers' seem to have a slightly different meaning because they also include people who are continually staying abroad and returning home.

Methods – 2.2 Data sources: climate indices

  • For the non-expert, differentiating between ENSO, the Southern Oscillation Index, the Oceanic Nino Index ONI, and Indian Ocean Dipole Mode Index is difficult. Can you add one or two sentences explaining how they are related?

Authors’ response:

  • The following sentences have been added.
  • Just as ENSO-related climatic fluctuations are presented using ONI indicators, the Indian Ocean Dipole Mode Index (DMI) is used to present the IOD-related climatic fluctuations. (Line 87-89)

  • ENSO is measured by ONI, and the IOD by DMI? You state that DMI is independent of ENSO, but why was it then included in the models? Also, in Figure 2A the two seem quite correlated in their occurrence.

Authors’ response:

  • Southeast Asia and South Asia, which are the main target regions for this study, are located between the Indian Ocean and the Pacific Ocean. Thus, it is affected by the climatic fluctuation in both the Indian Ocean and the climatic fluctuation in the Pacific Ocean. Therefore, we included both of those indices in the model.
  • ENSO and IOD are independent Pacific and Indian Ocean climatic fluctuation. However, since it belongs to a single planet, and the Pacific Ocean and the Indian Ocean are connected, synchronization could occur when strong fluctuations occur. This is the same reason as two seesaws facing each other are moving in the same direction at some points. More information on this can be found in references 7, 20 and 21.

Methods – 2.3 Data analyses

  • P 2, line 84 “we categorised cases by the countries suspected of causing infection in the imported cases during this period.” Unclear – what is suspected to cause the infections? A country cannot be a cause.

Authors’ response:

  • The mentioned sentence has been modified as follows.
  • After identifying trends in domestic and imported cases of shigellosis reported in South Korea from 2004 to 2017, we classified imported cases by the countries where transmission of the infection was estimated to have occurred. The country in which the transmission of such shigellosis infection is estimated to have occurred was based on the records of the epidemiological investigation. (Line 94-97)

  • 2 line 88 “dependent variable number of patients .... categorised by month” – means the number of monthly cases?

Authors’ response:

  • The mentioned sentence has been modified as follows.
  • The dependent variable is the number of monthly shigellosis patients reported to be infected during travel to South and Southeast Asia. (Line 100-101)

Results

  • P3, first paragraph of results: some sentences are unclear, please consider revising.
  • g. “total number of patients, which stood at 487 in 2004, has been reported to be within 110 patients”,

Authors’ response:

  • We have revised as follows.
  • The total number of cases, which occurred close to 500 cases in 2004, has decreased significantly since 2014 and is reported to be less than 120 cases. (Line 128-130)

  • Line 119 “Southeast Asia was the highest with 476 cases…” make clear these are the number of imported cases, not the total number in that region.

Authors’ response:

  • It has been revised to the following sentence.
  • When assessing the overseas regions suspected of originating imported infections, 476 cases (68.2 %) were estimated as being infected with shigellosis while travelling in Southeast Asian countries. The next frequent regions were; 132 cases (18.9 %) in South Asian countries, 63 cases (9.0 %) in East Asian countries, and 27 cases (3.8 %) in other regions. More specifically, in Southeast Asian countries, 137 cases (19.6%) were reported among those who travelled to Cambodia, followed by the Philippines with 119 cases (17.0%) and Vietnam with 89 cases (12.8%). A total of 132 cases were reported by travellers who visited South Asian countries, with 124 cases (17.8%) being infected during travel to India, accounting for most cases. In East Asia, it was the largest in China, with 48 cases (6.9%). (Line 132-140)

  • Figure 2 and text about outbound travellers might be better presented before showing the number of shigellosis cases among returning travellers (or in the appendix).

Authors’ response:

  • The number of outbound travellers is used because official statistic only presents the number of outbound travellers, not the number of returning travellers. Since the number of outbound travellers was used as the denominator when analyzing the data, the concept of incidence is included in the model. In the figure, rather than providing the incidence of shigellosis among returning travellers, we wanted to show the phenomenon that the number of outbound travellers is constantly increasing. This is because the increase in international tourism is an important variable in our research model.

  • P6 line 149 “La Nina” – unclear how this relates to the main independent variable ONI. I suggest to refer only to ONI in the results, and describe its relation to El Nino/La Nina patterns in the discussion. This will help readers without climatological expertise to follow the paper.

Authors’ response:

  • As you pointed out, the word 'La Niña' has been removed from this sentence and has been modified to be mentioned only in the discussion.

Discussion

  • 7/8 line “El Niño (La Niña) is based on a...”; P.8, line 162/163 “Nina (Nino) … 0.5°C or higher (-0.5 °C or lower)”. Summarizing the two phenomena in this way is efficient but difficult to follow for non-climatologists. I suggest splitting the effects of Nino/Nina in two separate sentences, or merging them in some other way.

Authors’ response:

  • The above sentence has been separated into two sentences.
  • The first month of the period when the ONI of 0.5°C or higher lasts more than five months is considered the starting point for El Niño. Additionally, the first month of the period when the ONI of 0.5°C or lower lasts more than five months is considered the starting point for La Niña. (Line 185-187)

  • Same sentence: “The first month is considered the start of El Niño” - unclear what is meant, at least for non-experts.

Authors’ response:

  • To clarify the meaning of the sentence, it has been modified to the same sentence as the answer to the previous question.

  • 8, line 178: Botswana is in Southern Africa, not in South Africa.

Authors’ response:

  • As you pointed out, we revised it to Southern Africa.

  • 8 line 185 and other places: I would avoid value-loaded terms like “developing / developed countries” (although even the UN uses these expressions). But all countries are developing (hopefully). Rather use “low- and middle-income countries, high-income countries”.

Authors’ response:

  • As you pointed out, we revised it to low- and middle-income countries

  • Line 166 “…indicators of events potential climate-related impact” – grammar / unclear what is meant

Authors’ response:

  • We revised it to the sentence below.
  • ENSO and IOD are climate fluctuations in the Pacific and Atlantic Oceans but are known as important indicators of events that can affect climate globally through teleconnection of the atmosphere and oceans.

  • Line 190-192: Good and important point. It might fit better at the end of the discussion or into the conclusion, maybe even in more detail.

Authors’ response:

  • The position of the sentence has not been changed due to the continuous flow of the contents, but we added more detailed contents like the sentence below to help to understand.
  • This is because travellers of high-income countries, with well-established legal infectious disease reporting systems, who travel and return to Southeast Asia and South Asia can be considered as random samplers as a proxy of residents of Southeast Asia and South Asia.

  • Line 197 “the largest number of imported shigellosis cases reported in 2010-2011 was related to the occurrence of the strongest La Nina” – that implies causality, but could be coincidence as it is just one observation.

Authors’ response:

  • The phrase 'related to' which implies the meaning of causality has been changed to 'associated with'.
  • the largest number of imported shigellosis cases reported in 2010-2011 was associated with the occurrence of the strongest La Niña at that time during the study period.

  • Line 203-205 “Therefore, when the fluctuation in the number of imported…” Unclear what is meant – country-specific analyses and results were not described before.

Authors’ response:

  • If the imported shigellosis cases are classified by each country visited by Korean travellers and compared with climate indices, it means that the association may not be clear due to the difference in various risk factors in each country. Separating the number of cases by country assigns too few cases, so country-specific analyses are not useful.
  • To help you understand this sentence more clearly, we have added the following sentence.
  • Also, it is already known that when El Niño/La Niña occurs, drought/flood frequently occurs in Southeast Asia and South Asia. However, in small regions or individual countries, geographic differences may cause weather events that differ from the global climatic phenomenon.

Conclusions

  • Good summary of paper. I would suggest to expand the conclusions – what do study finding imply beyond its scope? Possibly relevance in the context of future climate change and increasing tourism?

Authors’ response:

  • The following sentences have been added.

It is difficult to predict in advance how frequently large climatic events such as El Niño and La Niña will occur in the distant future. However, the number of international tourist arrivals in South and Southeast Asian countries is increasing rapidly every year. Therefore, it can be expected that cases of shigellosis will increase significantly among international tourists visiting this area during the La Niña period. Considering these characteristics, guidance is required to pay attention to hand hygiene and food consumption to tourists visiting the South and Southeast Asian countries during the La Niña period. (Line 288-294)

Round 2

Reviewer 3 Report

Dear authors,

thank you for the revised version of the manuscript.

Most of my comments have been addressed.

Here some minor issues:

  • Check grammar/clarity of revised / added text

  • Line 226 "This is because travellers of high-income countries, with well-established legal infectious disease reporting systems, who travel and return to Southeast Asia and South Asia can be considered as random samplers as a proxy of residents of Southeast Asia and South Asia."

    The sentence is a bit unclear - why are HIC travelers a good proxy for Asia LMIC residents? They are, obviously, due to similar exposure to local weather, water contamination etc, but this should be made clearer.

  • This point would also need to be linked in the conclusions, which are now only related to HIC travelers to the region:

    "It is recommended that guidance is required to pay attention to hand hygiene and food consumption to tourists visiting the South and Southeast Asian countries..."

  • Line 287 "It is difficult to predict in advance how frequently large climatic events such as El Niño and La Niña will occur in the distant future".
    There are some studies on climate change that asssess projected ENSO changes. I suggest to consider the latest IPCC report (WG 1) or newer publications as references.

Author Response

  • Line 226 "This is because travellers of high-income countries, with well-established legal infectious disease reporting systems, who travel and return to Southeast Asia and South Asia can be considered as random samplers as a proxy of residents of Southeast Asia and South Asia."

The sentence is a bit unclear - why are HIC travelers a good proxy for Asia LMIC residents? They are, obviously, due to similar exposure to local weather, water contamination etc, but this should be made clearer.

  • This point would also need to be linked in the conclusions, which are now only related to HIC travelers to the region:

"It is recommended that guidance is required to pay attention to hand hygiene and food consumption to tourists visiting the South and Southeast Asian countries..."

Authors’ response

  • The purpose of this study was to assess the impact of infectious disease outbreaks in relation to the climate events occurring across the wide range of geographic regions, using incoming report from travelers as a sentinel surveillance. It would be ideal to be able to obtain long-term time series data on shigellosis for all countries in the region, but there are inevitable gaps among countries in terms of its validity and consistency. Therefore, the authors are focusing our research on the report from incoming travelers rather than on local residents, which will assure the consistency and validity, while sacrificing some of the representativeness of the region and time.
  • This methodology has also been applied to studies on the occurrence of dengue fever in Japanese travelers who have traveled to Southeast Asia. (Fukusumi M. et al. Dengue Sentinel Traveler Surveillance: Monthly and Yearly Notification Trends among Japanese Travelers, 2006-2014. PLoS Negl Trop Dis. 2016. 19;10(8):e0004924.)
  • We have revised the discussion according to your suggestion (Line 222-229).

  • Line 287 "It is difficult to predict in advance how frequently large climatic events such as El Niño and La Niña will occur in the distant future".
    There are some studies on climate change that assess projected ENSO changes. I suggest to consider the latest IPCC report (WG 1) or newer publications as references.

Authors’ response

  • IPCC AR5 working group I report was reviewed and the report describes ENSO as follows.
  • "There is high confidence that the El Niño-Southern Oscillation (ENSO) will remain the dominant mode of interannual variability in the tropical Pacific, with global effects in the 21st century. Due to the increase in moisture availability, ENSO-related precipitation variability on regional scales will likely intensify. Natural variations of the amplitude and spatial pattern of ENSO are large and thus confidence in any specific projected change in ENSO and related regional phenomena for the 21st century remains low."
  • In consideration of the IPCC report's contents and the confidence level, it has been revised to the following.
  • It is predicted that extreme rainfall frequency will increase in the future due to climate change. Still, it is difficult to project in advance how the amplitude and spatial pattern of climatic events such as El Niño and La Niña will occur in the distant future. (Line 287-289)